https://doi.org/10.1038/s41467-019-12935-7` **OPEN**

# Procr-expressing progenitor cells are responsible for murine ovulatory rupture repair of ovarian surface epithelium

Jingqiang Wang[1], Daisong Wang[1], Kun Chu[2], Wen Li[2]* & Yi Arial Zeng 📍[1]*

Ovarian surface epithelium (OSE) undergoes recurring ovulatory rupture and repair. The OSE replenishing mechanism post ovulation remains unclear. Here we report that the expression of Protein C Receptor (Procr) marks a progenitor population in adult mice that is responsible for OSE repair post ovulation. Procr+ cells are the major cell source for OSE repair. The mechanism facilitating the rapid re-epithelialization is through the immediate expansion of Procr+ cells upon OSE rupture. Targeted ablation of Procr+ cells impedes the repairing process. Moreover, Procr+ cells displayed robust colony-formation capacity in culture, which we harnessed and established a long-term culture and expansion system of OSE cells. Finally, we show that Procr+ cells and previously reported Lgr5+ cells have distinct lineage tracing behavior in OSE homeostasis. Our study suggests that Procr marks progenitor cells that are critical for OSE ovulatory rupture and homeostasis, providing insight into how adult stem cells respond upon injury.

---

[1] State Key Laboratory of Cell Biology, CAS Center for Excellence in Molecular Cell Science, Shanghai Institute of Biochemistry and Cell Biology, Chinese Academy of Sciences, University of Chinese Academy of Sciences, Shanghai 200031, China. [2] Reproductive Medicine Center, Changzheng Hospital, Second Military Medical University, Shanghai 200003, China. *email: liwen@smmu.edu.cn; yzeng@sibcb.ac.cn

The ovary is covered by a single-layer of ovarian surface epithelium (OSE), expressing Keratin 7, 8, 18, and 19[1–3]. Like most epithelia, the OSE requires replacement throughout the lifetime of the individual. During adult reproductive cycles, the OSE undergoes cyclic changes in parallel to ovary follicle growth[4,5]. A surge of the gonadotropin-releasing hormone and luteinizing hormone induces ovulation, which is associated with extensive architectural remodeling of the follicular structure, extrusion of the ovum, and the OSE rupture when releasing the oocyte[6]. After ovulation, the ruptured OSE is repaired. Considering the short estrus cycle (every 4–5 days) in mouse, the repair mechanism of the OSE is highly robust and efficient, with complete closure of the wound achieved within 12 h to 3 days following rupture[7,8]. This recurring process suggests the existence of stem/progenitor cells in the OSE. It is hypothesized that the rupture is replenished by stem/progenitor cell replication and migration from the wound edges[4,9]. However, little is known about the OSE-replenishing mechanisms during and after ovulation.

Earlier reports have identified a subset of epithelial cells that exhibit certain stem cell characteristics using label-retention or side-population techniques[9,10]. The side-population-enriched OSE cells, which also feature Stem Cell Antigen-1 (Sca1) expression, exhibit higher in vitro colony-formation capacity compared with other OSE cells[9]. More recently, two in vivo studies have identified Lgr5 as a stem/progenitor cell marker in OSE and oviduct using lineage tracing methods[11,12]. Although one study emphasizes that these stem cells are restricted to the adult ovary hilum[11], a region hinging the ovary and fimbria, which has been described as the stem cell niche[11,13], the other demonstrates that Lgr5+ cells are distributed throughout the OSE[12]. The latter study highlights the robust role of Lgr5+ cells during embryonic and neonatal ovary development. Yet, the contribution of Lgr5+ cells in adult OSE homeostasis is relatively modest, with formation of small clones in adult lineage tracing experiments[12]. It is possible that another progenitor population exists in adult OSE and is responsible for homeostasis and repair.

Previous studies from us and others have reported Protein C receptor (Procr) marks stem cells in several adult tissues, including the mammary gland[14], endothelium[15], and hematopoietic systems[16–19]. Procr is a single-pass transmembrane protein[20], with established roles in anticoagulation and cytoprotection[21]. In this study, we investigate the role of Procr in OSE homeostasis and post rupture repair. We generate a Procr-rtTA knock-in allele to scrutinize Procr+ cell's distribution and behavior. We conduct meticulous in vivo lineage tracing and targeted cell ablation during individual estrus cycles using the Procr-CreER model. We establish an ovary whole-mount imaging technique for the comprehensive evaluation of the Procr+ cell contribution to OSE repair. Finally, we compare the contribution of Procr+ cells and Lgr5+ cells in adult OSE homeostasis.

## Results

**A small population of OSE cells features Procr expression**. To probe the expression of Procr in the OSE, we performed RNA in situ hybridization. We found that a subset of OSE cells express Procr, including those in the hilum and other OSE areas (Fig. 1a, b). The overall proportion of Procr+ cells was $10.40 \pm 0.54\%$ of total OSE cells (Fig. 1c). To better visualize the pattern of Procr+ OSE cells, we generated a Procr-rtTA knock-in mice, with rtTA inserted at the ATG start codon (Supplementary Fig. 1a–d). Subsequently, Procr-rtTA;tetO-H2B-GFP mice were generated by genetic crosses that result in the stable expression of histone 2B-GFP fusion protein in Procr+ cells after doxycycline (DOX) treatment. DOX was fed to 8-week-old adult

mice for 2 days, and ovaries were harvested. By immunostaining, we found that nuclear GFP+ cells were distributed across the OSE, including the hilum region, near the follicles, inter-follicles, and the corpus luteum, and next to the ovulatory rupture (Fig. 1d). It is noteworthy that H2B-GFP+ cells are also seen in endothelial cells surrounding or penetrating the follicles, consistent with the known Procr expression in endothelium and mesenchymal cell compartments[14,15] (Fig. 1d).

Next, we set to quantify Procr+ OSE cells precisely by fluorescence-activated cell sorting (FACS) analysis. At present, there is no reliable isolation method for mouse OSE. OSE cells, enriched using previous protocols, are not a pure population and contain non-OSE cell contamination[11,22]. By immunostaining, we found that OSE cells express EpCAM (Supplementary Fig. 2a). This enabled us to FACS-isolate OSE cells by surface EpCAM expression. OSE cells isolated by Lin–, EpCAM+ accounted for ~ 1% of all ovary cells (Supplementary Fig. 2b). The isolated OSE cell identity was validated by Krt8 (K8) and Krt19 (K19) expression by both quantitative PCR (qPCR) and immunostaining (Supplementary Fig. 2c–d). Further FACS analysis indicated that Procr+ cells comprise $9.78 \pm 0.84\%$ of OSE cells (Fig. 1e), consistent with the observation made using Procr RNA in situ and using GFP+ reporter cells in Procr-rtTA;tetO-H2B-GFP mice that Procr marks a small subpopulation of OSE cells.

**Procr+ cells contribute to OSE repair after ovulatory rupture**. To investigate the role of Procr+ cells during ovary development and homeostasis, we performed in vivo lineage tracing. We utilized Procr-CreER;R26-mTmG mice[14,23], and examined the contribution of Procr+ cells in various developmental stages. First, we focused on OSE repair after ovulatory rupture. A single dose of tamoxifen (2 mg/25 g body weight) was administered to 4-week-old mice to induce lineage tracing, and 2 days later, superovulation was induced by injecting PMSG (Pregnant mare's serum gonadotropin) and HCG (Human chorionic gonadotropin). Ovaries were harvested at 4 days post TAM induction (4d pi, pre-ovulation stage), 4.5d pi (ovulation, rupture), and 7d pi (wound repair completed), and GFP+ cells (progeny of Procr+ cells) were examined (Fig. 2a). To clearly see the distribution of GFP+ cells, we developed ovary whole-mount imaging (see methods). There were no GFP+ cells in the ovary of Procr-CreER; R26-mTmG mice without tamoxifen injection.

At the pre-ovulation stage (4d pi), we paid particular attention to the OSE near the large (ovulating) follicles that are maturing and will soon be released because the OSE rupture is thought to be replenished by local cells from the wound edges. Whole-mount imaging illustrated the dispersed location of GFP+ OSE cells (Fig. 2b), and section imaging showed small GFP+ clones residing on both sides of the pre-ovulation follicle (Fig. 2b, c). Strikingly, at ovulation stage (4.5 days pi), whole-mount imaging elegantly illustrated a zone of GFP+ cells surrounding the ruptured site (Fig. 2d). This suggested that during this narrow time window (0.5d), Procr+ cells have rapidly generated many progeny cells marked by GFP expression at the wound edge. Consistently, in section imaging, large GFP+ patches were readily detected on both sides of the rupture (Fig. 2e). GFP+ cells located near ruptured sites and other regions were quantified using section images. Significantly, more GFP+ cells were seen near ruptured sites compared with those of other regions, e.g., near small follicles, large follicles, and inter follicular region (Fig. 2f).

At 7d pi, in parallel with formation of the corpus luteum (C.L.) post ovulation, wound repair is completed. In whole-mount imaging, we observed big patches of GFP+ cells in the OSE covering the newly formed corpus luteum (Fig. 2g), likely a fusion

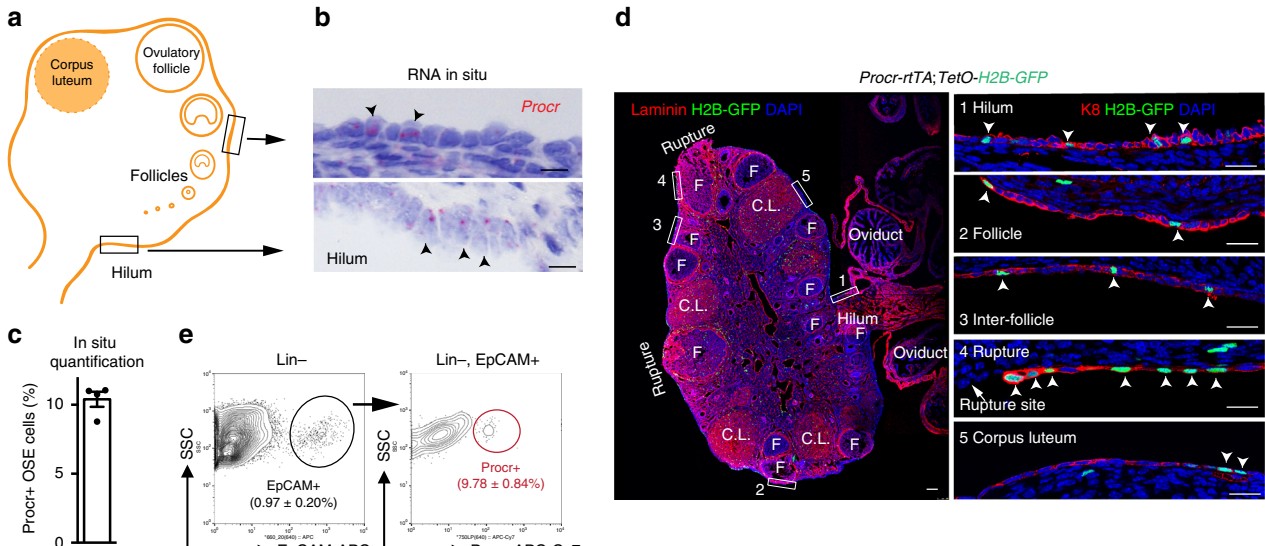

**Fig. 1** A subpopulation of OSE cells express Procr. **a** A schematic diagram of the mouse ovary with various stages of follicles. **b** RNA in situ indicating the expression of *Procr* in the adult mouse OSE, in both the non-hilum region (upper panel) and hilum region (lower panel). $n = 4$ mice and 16 images. Scale bar, 10 μm. **c** Procr+ cells percentage in the OSE based on RNA in situ results. Data are pooled from four independent experiments and presented as mean ±s.e.m. **d** *Procr-rtTA;TetO-H2B-GFP* mice were fed with doxycycline for 2 days. Confocal image of an ovary section showing the expression of Procr by histone 2B-GFP (arrowheads) in different regions of the OSE (F, follicle; C.L., corpus luteum). $n = 4$ mice and 16 images. Scale bars, 100 μm in full ovary view and 20 μm in zoom in view. **e** FACS analysis of adult mice OSE with EpCAM and Procr. The percentages of total OSE cells (Lin−, EpCAM+) and Procr + OSE cells (Lin−, EpCAM+, Procr+) are as indicated. Data are pooled from four independent experiments and presented as mean±s.e.m

of multiple clones. Significantly more GFP+ cells were located near the corpus luteum, compared with other regions shown by section imaging and quantification (Fig. 2h, i). Quantification of GFP+ cells indicated the continuous expansion of GFP+ progeny cells during pre-ovulation, ovulation, and through the final stage of wound repair (Fig. 2j). Together, these results suggest that Procr+ cells make a major contribution to OSE repair after ovulatory rupture.

**Procr+ cells contribute to OSE homeostasis in long-term.** We next investigate the contribution of Procr+ cells in adult homeostasis under physiological conditions. Tamoxifen was administered to 8-week-old adult *Procr-CreER;R26-mTmG* mice, and ovaries were harvested at 2 days (short-term), 4 weeks, and 4 months pi (long-term) (Supplementary Fig. 3a). In short-term tracing experiments, dispersedly individual GFP+ cells were seen in all regions across OSE (Supplementary Fig. 3b). After 4 weeks of tracing, the GFP+ cells expanded in numbers (Supplementary Fig. 3c). The OSE area covered by GFP+ cells continued to expand when examined after 4 months of tracing (Supplementary Fig. 3d–e). FACS analysis confirmed the increased percentage of GFP+ OSE cells along the tracing period (Supplementary Fig. 3f–g). These data suggest that Procr+ cells make a significant contribution during adult OSE homeostasis.

Tracing experiments conducted at various developmental stages delivered similar results. In pubertal mice (4-week-old), Procr+ cells were traced for different time periods: 2 days, 4 weeks, 5 months, and 14 months (Supplementary Fig. 4a). Quantification revealed increasing number of progeny cells over time (Supplementary Fig. 4b–f). The prevalence of big clones after long-term tracing suggests that Procr+ cells self-renew and are long-lived (Supplementary Fig. 4e–f). Together, these in vivo lineage tracing experiments indicate that Procr+ cells are enriched for adult OSE progenitor cells.

**Procr+ cells expand instantly upon ovulation.** Procr+ cells rapidly gave rise to progeny cells during ovulatory repair and the progeny's contribution to the process seemed dominant; thus we further investigated the behavior of Procr+ cells themselves. Interestingly, we found that the percentage of Procr+ OSE cells (Lin− EpCAM+, Procr+) quickly increases within 0.5 day from pre-ovulation (8.7 ± 0.4%) to ovulation (20.1 ± 0.7%), followed by a decline to 9.7% ± 0.4% when repair is complete (Fig. 3a–c).

To locate and visualize the expansion of Procr+ cells, we utilized *Procr-rtTA;TetO-H2B-GFP* mice. 4 weeks old mice were fed with DOX continuously for 7 days, superovulation was induced at 2d pi, and ovaries were analyzed at 4d pi (pre-ovulation), 4.5d pi (ovulation), 7d pi (completed repair) (Fig. 3d). Under this continuous DOX-feeding strategy, not only were the existing Procr+ cells labeled with nuclear GFP, but also can peak GFP levels be maintained in the descendant cell that remains Procr+. Confocal images were stacked to include all potential GFP signals in a given cell, for both Procr+ cells and their progeny cells. At 4d pi (pre-ovulation), we observed 6.65 ± 0.31 cells with the peak GFP level on the OSE covering the pre-ovulating follicle (~ 40–50 DAPI+ cells in the scored area) (Fig. 3e, f). At 4.5d pi (ovulation), more GFP+ OSE cells were seen on both edges of the ruptured site, with 13.26 ± 0.58 cells expressing the peak level of GFP at the wound edge (Fig. 3f, g). These results were consistent with the twofold expansion of Procr + cells obtained by FACS, and further located the expansion of Procr+ OSE cells to both sides of the wound edge. At 7d pi, when OSE repair was complete, we observed that while the peak GFP signal intensity have decreased, the number of total GFP+ cells further expanded (Fig. 3h, i) at the OSE of newly formed corpus luteum. These results suggest that divisions from 4.5 to 7d pi have generated many Procr− progeny cells that have inherited one half (one division) of or less (two or more divisions) GFP intensity. The step-wise dilution of GFP signal intensity was confirmed by FACS analysis. At 4d pi and 4.5d pi, most majority cells GFP+

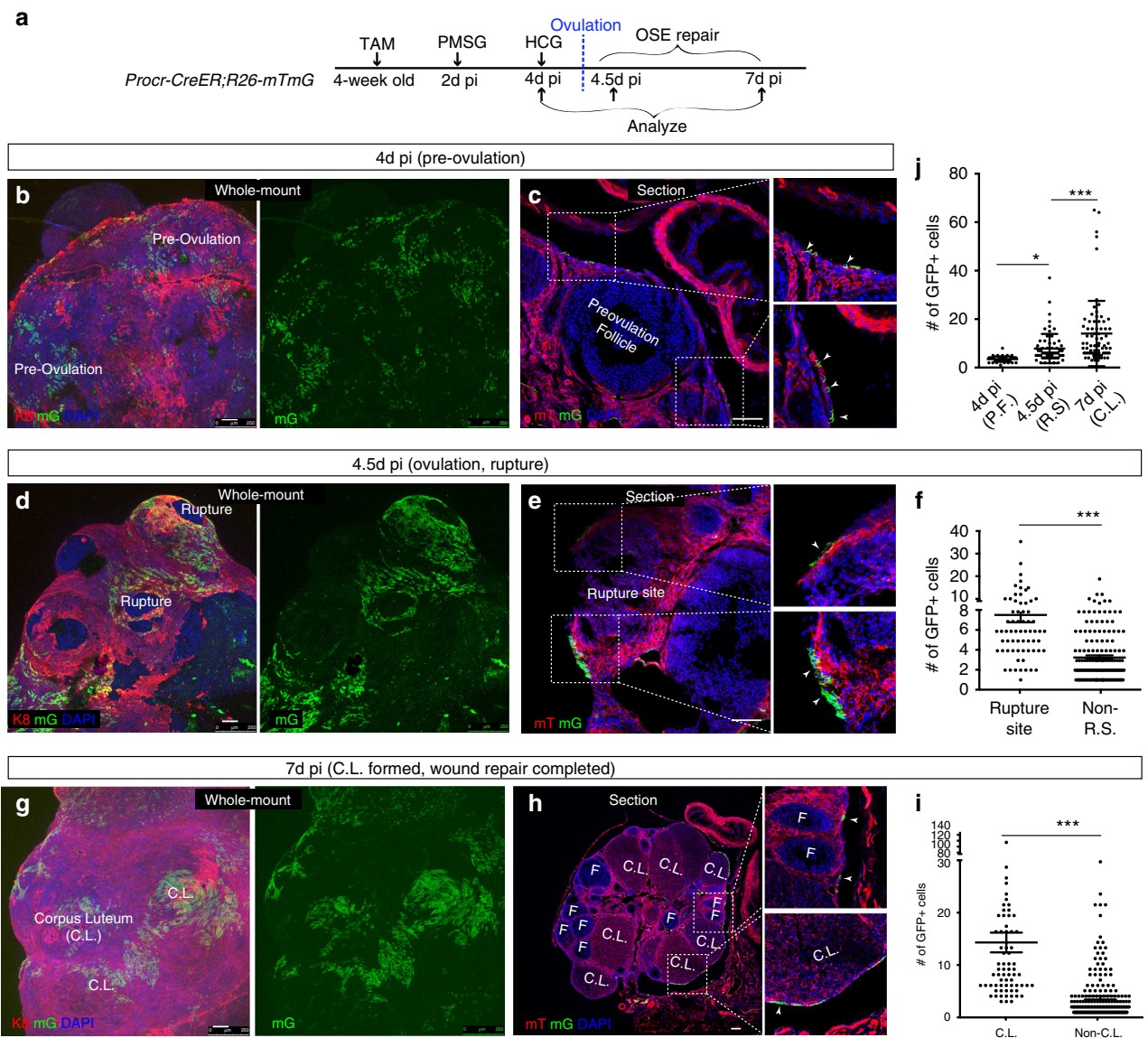

**Fig. 2** Procr+ cells contribute to OSE repair after ovulation. **a** Illustration of lineage tracing and superovulation-inducing strategy using *Procr-CreER;Rosa26-mTmG* mice. **b**, **c** At 4pi (pre-ovulation stage), ovary whole-mount confocal imaging showing dispersed GFP+ cells in the OSE (**b**). n = 4 mice and eight ovaries images. Section imaging confirming the appearance of a few GFP+ OSE cells on both sides of the pre-ovulation follicle (**c**). Scale bars, 100 μm. n = 4 mice and 16 images. **d**–**f** At 4.5pi (ovulation stage), ovary whole-mount confocal image demonstrating zones of concentrated GFP+ cells surrounding the rupture site (R.S.) (**d**). Section imaging confirming the appearance of abundant GFP+ OSE cells at both edges of the rupture site (**e**). Scale bars, 100 μm. Quantification of GFP+ cell numbers (per 0.04 mm² area boxed by the dashed line) showing that GFP+ cells by rupture sites are significantly more than those in randomly selected non-R.S. (**f**). n = 4 mice, > 40 areas were counted in each group. Unpaired two-tailed t test is used for comparison. ***P < 0.001. **g**–**i** At 7pi (repair completed stage), ovary whole-mount confocal imaging showing large GFP+ patches located at corpus luteum (C.L.) area that can be distinguished by the residual protrusion (**g**). Section imaging demonstrating large patch of GFP+ cells in the OSE covering corpus luteum (C.L.), whereas GFP+ cells are fewer in areas where the OSE covers the follicle (F) (**h**). Scale bars, 100 μm. Quantification of GFP+ cells (per 0.04 mm² area) showing that GFP+ cells in the C.L. area are significantly more than those in randomly selected non-C.L. sites (**i**). n = 4 mice, > 40 areas were counted in each group. Unpaired two-tailed t test is used for comparison. ***P < 0.001. **j** Quantification of GFP+ cell numbers showing the continuous expansion of GFP+ cells during tracing. At 4d pi, GFP+ cells in the pre-ovulation follicle (P.F.) area were counted. At 4.5d pi, GFP+ cells flanking the rupture sites (R.S.) were counted. At 7d pi, GFP+ cells covering the C.L area were counted. Each view was 0.04 mm² in size. n = 4 mice, > 40 areas were counted for each time point. One-way ANOVA with Tukey test is used for comparison of multiple groups. *P < 0.05, ***P < 0.001

cells were in peak GFP intensity, whereas at 7d pi, GFP signal intensity had decreased (Supplementary Fig. 5a–b).

The prompt expansion for Procr+ cell pool is intriguing. One possibility is that Procr− cells turn into Procr+ cells upon ovulatory rupture. Another possibility is that Procr+ cells symmetrically divide to double the pool. The latter scenario involves rapid proliferation of Procr+ cells. Thus, we compared the proliferative capacity of Procr+ (Lin−, EpCAM+, Procr+)

and Procr− (Lin−, EpCAM+, Procr−) cells after 10 h EdU incorporation. In non-stimulated non-staged (for estrus cycle) ovaries, Procr+ cells exhibited 2.9-fold higher EdU incorporation compared with Procr− cells (Fig. 3j). Upon PMSG + HCG induced superovulation, Procr+ cells displayed 10.9-fold higher EdU incorporation compared to Procr− cells (Fig. 3j). Comparing Procr+ cells at superovulated state and those at non-stimulated state, their proliferative capacity further enhanced,

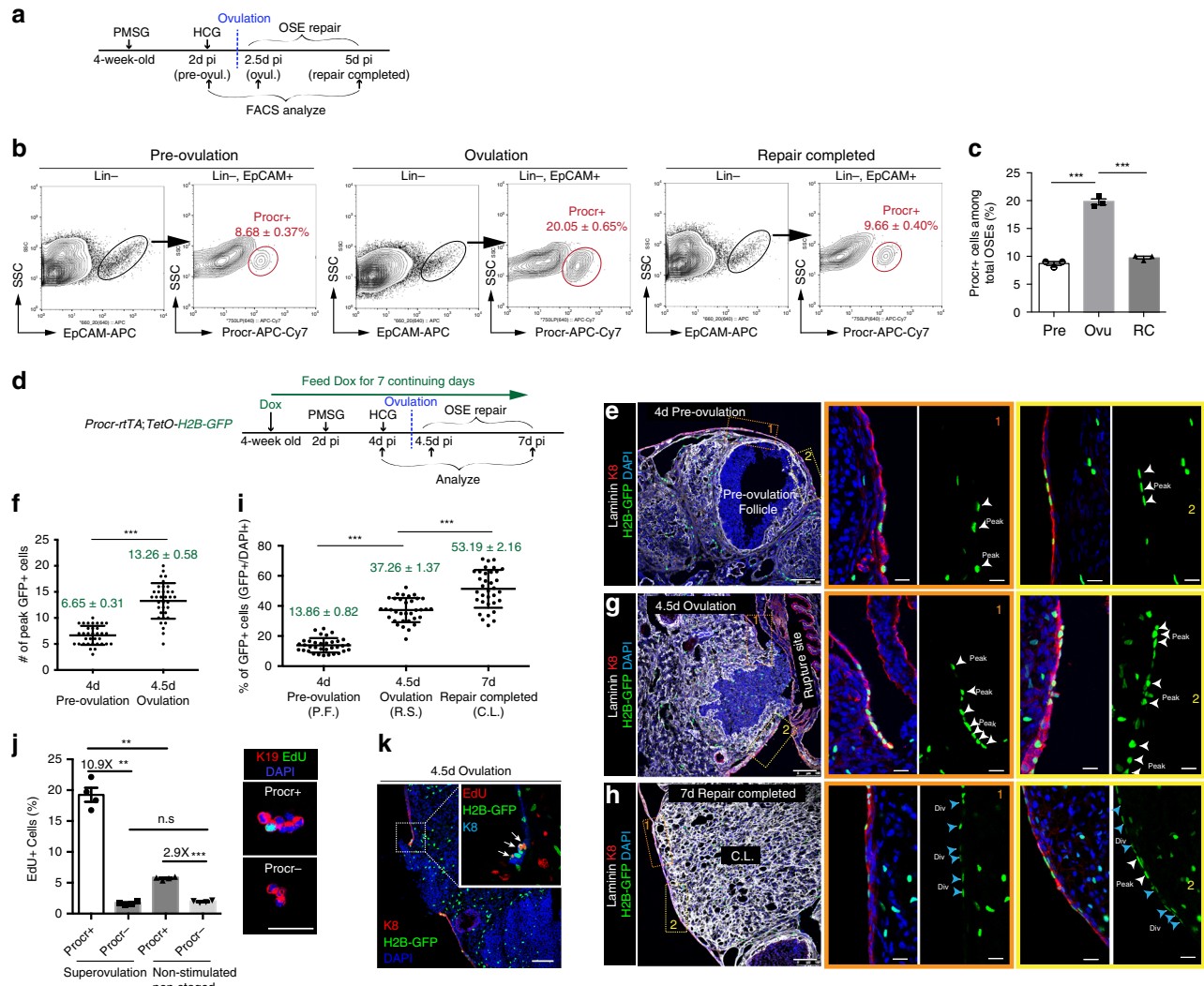

**Fig. 3** Procr+ cells expand immediately upon ovulation. **a** Illustration of superovulation and analysis strategy as indicated using wildtype mice. **b, c** FACS analysis (**b**) and quantification (**c**) of the percentage of Procr+ OSE cells at three different stages as indicated. $n = 3$ mice. One-way ANOVA with Tukey test is used for comparison of multiple groups. ***$p < 0.001$. **d** Illustration of the visualization and tracing strategy for Procr+ cells using *Procr-rtTA;TetO-H2B-GFP* mice. **e, f** At 4 days after Dox feeding (pre-ovulation), confocal images showing peak level of nuclear GFP+ cells in the OSE close to pre-ovulation follicle. Scale bars, 100 μm. $n = 3$ mice and 15 images. More than 30 pre-ovulation follicles were scored. **f, g** At 4.5d pi (ovulation), confocal images showing increasing numbers of peak levels of nuclear GFP+ cells in the OSE on both edges of the rupture. Scale bars, 100 μm. $n = 3$ mice and 15 images. More than 30 rupture sites were scored. Data are presented as mean±s.e.m. Unpaired two-tailed $t$ test is used for comparison. ***$p < 0.001$. **h** At 7d pi (repair completed), confocal images showing GFP+ cells with various diluted signal intensities in the OSE covering the newly formed corpus luteum. Scale bars, 100 μm. $n = 3$ mice and 15 images. More than 30 corpus luteum were scored. **i** Percentages of GFP+ cells of all levels in the OSE covering pre-ovulation follicle, ovulating follicle, and corpus luteum were quantified, showing the increasing number of GFP+ cells over time. $n = 3$ mice in each time point. Data are presented as mean±s.e.m. One-way ANOVA with Tukey test is used for comparison of multiple groups. ***$p < 0.001$. **j** After 10 hrs EdU incorporation, Procr+ and Procr− cells were FACS-isolated and underwent cytospin and EdU and K19 staining. Representative images are shown on the right, quantification of EdU+ cells is shown on the left. $n = 3$ mice in each time point. Data are presented as mean±s.e.m. One-way ANOVA with Tukey test is used for comparison of multiple groups. **$p < 0.01$, ***$p < 0.001$. **k** At 4.5d pi (ovulation), confocal images of *Procr-rtTA;TetO-H2B-GFP* ovary sections showing EdU staining in GFP+ (Procr+) cells at the edges of ruptured site. Scale bars, 100 μm. $n = 3$ mice and 15 images

whereas the proliferation of Procr− cells appeared not affected by the increased events of ovulatory rupture (Fig. 3j). Consistently, immunostaining of *Procr-rtTA;TetO-H2B-GFP* ovary sections showed that at the edges of ruptured site, GFP+ cells have high incidence of EdU staining (Fig. 3k). Collectively, these results suggest that Procr+ cells have higher proliferative capacity than Procr− cells, and the proliferation of Procr+ cells is further increased upon ovulatory rupture.

Results from lineage tracing experiments (Fig. 2) also favor the model that Procr+ cells symmetrically divide to double the progenitor pool. Given that TAM was administered at 4 days

before the rupture, and TAM-induced recombination events are most active for a short time period (~ 2 days)[24–26], presumably only existing Procr+ cells were labeled before ovulation. Upon rupture, if Procr− cells converted to Procr+ cells, these de novo Procr+ cells would not be labeled and would not be detected as GFP+ cells in the expansion documented in Fig. 2.

The lineage tracing behavior of Procr+ cells was further interrogated using a DOX-mediated pulse and chase strategy. 4 weeks old *Procr-rtTA;TetO-H2B-GFP* mice were fed with DOX for 2 days, at which time the existing Procr+ cells were labeled with nuclear GFP. This was followed by superovulation and 5-day

chase period after cessation of DOX (Supplementary Fig. 6a). At 4d pi (pre-ovulation), we observed 13.68 ± 0.78% of GFP+ cells on the OSE near the pre-ovulating follicle (Supplementary Fig. 6b, e). At 4.5d pi (ovulation), increasing numbers of GFP+ OSE cells were seen on both edges of the ruptured site (32.60 ± 1.58%) (Supplementary Fig. 6c, e). At 7d pi, when OSE repair was completed, GFP+ OSE cell numbers had further expanded (44.31 ± 2.51%) (Supplementary Fig. 6d, e). These were consistent with the lineage tracing results, showing that Procr+ cells and their descendant cells contribute towards ovulatory repair.

Together, these data suggest that in response to ovulatory rupture, Procr+ cells by the rupture edge symmetrically divide to expand the progenitor cell pool, followed by rounds of division to generate Procr− progeny cells. The net outcome, as shown by FACS analysis, was expansion of Procr+ cells upon OSE rupture; when repair is completed, Procr+ cells return to their original numbers that are required for their role in maintaining homeostasis.

**Robust clonogenicity and in vitro expansion of Procr+ cells.** In vitro colony formation has been implicated as one of stem/progenitor cell characteristics. To investigate the colony-forming ability of Procr+ cells, as illustrated in Fig. 4a, total OSE cells (Lin−, EpCAM+), Procr+ cells (Lin−, EpCAM+, Procr+) and Procr- cells (Lin-, EpCAM+, Procr–) were isolated and cultured in three-dimensional (3D) Matrigel as previously described[9,12]. By culture day 7, both total OSE and Procr+ cell groups were able to form colonies with sizes ~40 μm in diameter (Fig. 4b). Procr+ cells exhibited sixfold higher colony-formation efficiency compared with total OSE cells, with about one colony formed out of 10 plated Procr+ OSE cells (Fig. 4c). In contrast, Procr− cells were not able to form colony, suggesting that cells with colony-forming ability have been depleted from this group (Fig. 4b, c).

It has been reported that a subpopulation of OSE cells express Sca1 and is enriched for sphere/colony-forming ability[9]. Therefore, we examined the relationship of Sca1+ cells and Procr+ cells in the OSE. Using the updated FACS analysis protocol described in Supplementary Fig. 2, we found that Sca1+ cells comprise 11.18 ± 1.51% of OSE cells (Fig. 4d). Interestingly, Sca1+ cells were largely overlapping with Procr+ cells, and all Procr+ cells were Sca1+ (Fig. 4e). These results are in line with the enhanced colony-formation ability seen in Procr+ OSE cells.

Next, in an attempt to passage the colonies arisen from Procr+ cells, day 7 colonies were dissociated to single cells and replated in 3D culture. Although 2nd colonies were able to form, colony numbers did not expand, suggesting that the numbers of progenitor cells did not increase in the primary culture. We also noted that Procr expression was poorly maintained in the primary colonies, with Procr+ cells indicated by nuclear GFP expression using Procr-rtTA;TetO-H2B-GFP mice (1.26 ± 0.17 cells with bright GFP signal per colony) (Fig. 4f, h). As Procr is a Wnt target[14], we supplemented Wnt3a in the culture. Procr+ cells were readily detected in colonies (4.96 ± 0.48 cells with bright GFP signal per colony) in the presence of Wnt3a (Fig. 4g, h). The increased Procr level was confirmed by qPCR analysis (Fig. 4i). With the addition of Wnt agonist CHIR99021, colonies arisen from Procr+ cells can be serially passaged for more than seven times and for more than 2 months, with the colony number expanded for five- to sixfolds in each passage (Fig. 4j). K8 staining verified that their OSE cell identities were maintained in the primary and passaged colonies (Fig. 4k). Together, these results suggest that Procr+ OSE cells have robust clonogenicity, and they can be propagated in culture for long-term in the presence of Wnt signals.

**Procr+ cells are essential for OSE homeostasis and repair.** To evaluate the significance of Procr+ cells in OSE homeostasis and ovulatory rupture repair, we performed targeted ablation of these cells using diphtheria toxin (DTA). For the ablation, TAM was administered to Procr-CreER;R26-DTA mice (4-week-old), followed by the superovulation procedure. We examined the impact of losing Procr+ cells on OSE repair. Ovaries were harvest at 7d pi, when the corpus luteum had formed, and OSE repair is expected to be complete (Fig. 5a). The same procedure was carried out on control Procr-CreER mice. When analyzed at 7d pi, the intact OSE, as seen by K8 staining as well as ovary whole-mount imaging, clearly showed protrusions at what had been the ovulatory rupture site (Fig. 5b). The integrity of OSE covering the newly formed corpus luteum was confirmed in confocal sections (Fig. 5c). In contrast, when Procr-CreER;R26-DTA mice were analyzed at 7d pi, OSE cells were missing at sites of ovulation as shown by lack of K8-expressing cells in the whole-mount imaging (Fig. 5d). Consistently in sections, gaps were seen in the OSE layer, especially at corpus luteum area (Fig. 5e). These results suggest that the OSE repair process was interrupted when Procr+ cells were ablated.

We also evaluated the impact of Procr+ cell ablation during adult homeostasis under physiological conditions. Procr-CreER;R26-DTA adult mice were administered with tamoxifen, and OSE phenotypes were analyzed after 1 month, a timespan of 5–6 natural estrus cycles (Supplementary Fig. 7a). FACS analysis showed the percentage of Procr+ OSE cells was reduced from 9.22 ± 0.38% in control to 2.26 ± 0.27% in DTA group, confirming the successful ablation (Supplementary Fig. 7b–c). The control group displayed an intact OSE layer (Supplementary Fig. 7d), whereas ablation of Procr+ cells resulted in loss of OSE cells (Supplementary Fig. 7e). Together, these results suggest that Procr+ cells are essential for OSE homeostasis and repair.

**Procr+ and Lgr5+ cells exhibit distinct tracing behaviors.** Next we compared Procr+ cells with previously identified Lgr5+ OSE stem/progenitor cells[11,12]. Using Lgr5-CreER-IRES-EGFP mouse as a reporter[27], combined with the FACS protocol developed in this study, we found that Lgr5+ (GFP+) cells comprised 40.6 ± 4.2% of total OSE cells, and were not overlapping with Procr+ OSE cells (Fig. 6a). We compared these two populations by in vivo lineage tracing experiments, with the same reporter line (R26-tdTomato) and the same induction condition. In agreement with the previous report[12], clones generated in adult mice by Lgr5+ cells were small after 1 month of tracing (5–6 estrus cycles) (Fig. 6b). The pattern of small, yet frequent, clones was consistent with the high percentage of Lgr5+ cells seen in the OSE. In contrast, progeny of Procr+ cells were distributed in patches covering larger areas of the OSE within 1 month of tracing, as shown by whole-mount (Fig. 6c); this likely indicates a merger of multiple big clones. Importantly, labeled cells were quantified by FACS and this analysis confirmed a significantly greater expansion of Procr-derived progeny (3.1-fold increase) compared with that of Lgr5-derived progeny (no significant change) (Fig. 6d). These results suggest that Procr+ OSE cells and Lgr5+ OSE cells are distinct populations, and Procr+ cells exhibited a more significant contribution to adult OSE homeostasis under physiological conditions.

It has been reported in other organs (e.g., the small intestine) that different stem/progenitor cell populations exist, and ablation of one can be rescued by another[28,29]. Thus, we test whether injuries, induced by loss of Procr+ cells, can be compensated by Lgr5+ cells. To this end, we generated Procr-rtTA;TetO-DTA;Lgr5-CreER;R26-tdTomato mice, in which Procr+ cells can be ablated by Dox induction, and simultaneously Lgr5+ cell

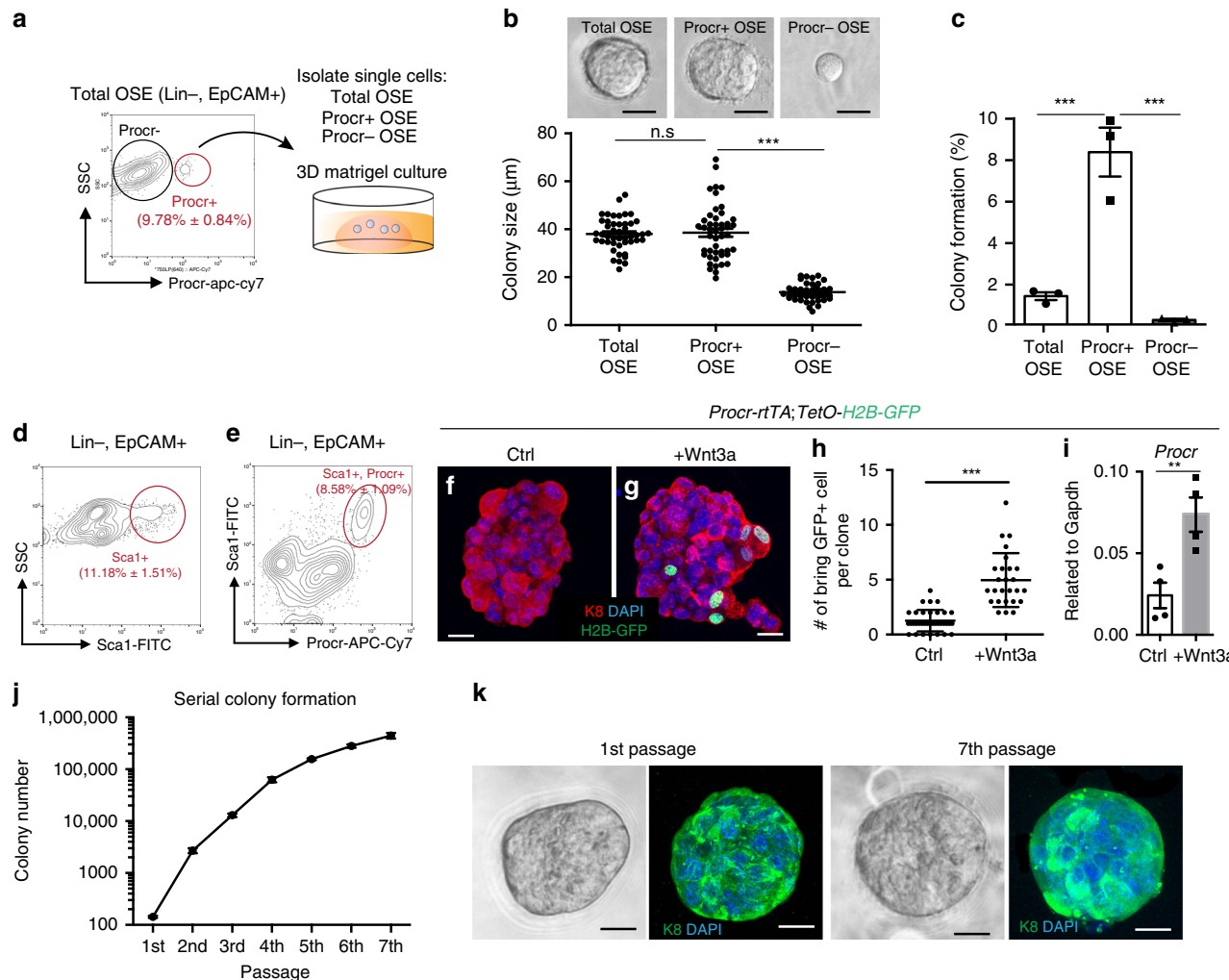

**Fig. 4** Procr+ cells have robust clonogenicity and can be expanded in vitro for long-term. **a** Illustration of OSE cell isolation and 3D Matrigel culture scheme. Total OSE cells (Lin−, EpCAM+), Procr+ OSE cells (Lin−, EpCAM+, Procr+) and Procr− OSE cells (Lin−, EpCAM+, Procr−) were FACS-isolated and resuspended in matrigel for 3D culture. **b** At day 7 culture, colony sizes were measured by the diameter in each group. Representative images were shown from 15 images. Data are pooled from three independent experiments and displayed as mean±s.e.m. Scale bars, 20 μm. One-way ANOVA with Tukey test is used for comparison of multiple groups. ***P < 0.001. n.s, not significant. **c** Colony numbers were counted and colony-formation efficiencies in each group were shown. Data are pooled from three independent experiments and displayed as mean±s.e.m. One-way ANOVA with Tukey test is used for comparison of multiple groups. ***P < 0.001. **d, e** FACS analysis of adult mice OSE with Sca1 and Procr. The percentages of Sca1+ OSE cells (Lin−, EpCAM+, Sca1+) **d**, and Sca1+, Procr+ OSE cells (Lin−, EpCAM+, Sca1+, Procr+) are as indicated. Data are pooled from three independent experiments and presented as mean±s.e.m. **f–h** Colonies were formed from OSE cells isolated from Procr-rtTA;TetO-H2B-GFP mice, in the presence of vehicle control (**f**) or Wnt3a (**g**). Doxycycline was added to the culture in the last 2 days before harvest, representative images were shown from 15 images. Number of bright GFP+ cells per colony were quantified (**h**). One of three similar experiments is shown, n = 3 mice. Scale bars, 20 μm. Unpaired two-tailed t test is used for comparison. ***P < 0.001. **i** qPCR analysis indicating the upregulation of Procr expression in OSE colonies in the presence of Wnt3a. Data are pooled from four independent experiments and displayed as mean±s.e.m. Unpaired two-tailed t test is used for comparison. **P < 0.01. **j** Colony numbers in serial passages were counted, showing that colony numbers had expanded in each passages. Data are pooled from two independent experiments and displayed as mean±s.e.m. **k** Representative images of 1st or 7th passage colonies from 15 images. K8 staining indicating the OSE property was maintained. Scale bars, 20 μm

activities can be traced by TAM administration. Adult mice were fed with Dox for 7 days to ablate Procr+ cells, followed by TAM administration to initiate Lgr5+ cell tracing; the OSE was examined in 1 month after TAM induction (Fig. 6e). Consistent with the experiments in Fig. S7 (ablation using Procr-CreER;R26-DTA model), ablation using Procr-rtTA;TetO-DTA led to loss of OSE cells as shown by loss of K8-expressing cells (Fig. 6e). Compared with control, which had no cell ablation, loss of Procr+ cells did not stimulate more active Lgr5+ cells division as shown by dispersed clones generated by Lgr5+ cells (Fig. 6e, f). These results suggest the distinct roles of Procr+ cells and Lgr5+

cells arise during adult OSE homeostasis, underscoring the notion that Procr+ cells are indispensable.

## Discussion

In this study, we report a population of Procr+ OSE progenitor cells that play a crucial role in homeostasis and ovulatory rupture repair in adult mice. Procr+ OSE cells constitute 10% of total OSE cells, residing in regions across the ovary surface (Fig. 7). The dominant contribution of Procr+ cells during ovulatory rupture repair is demonstrated by lineage tracing experiments

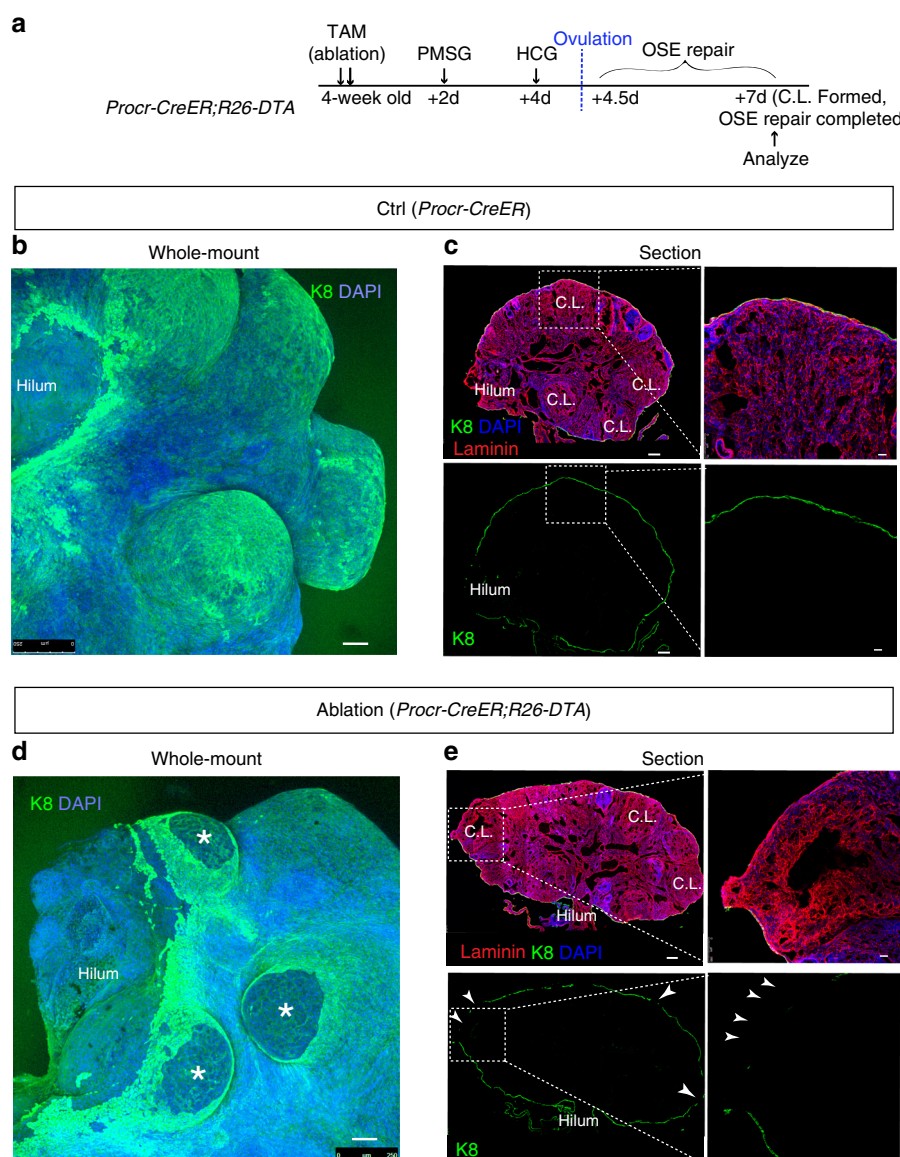

**Fig. 5** Procr+ cells are essential for OSE repair after ovulatory rupture. **a** Illustration of superovulation and the Procr+ cell-targeted ablation strategy. TAM was administered to 4-week-old *Procr-CreER;R26-DTA* mice, following by PMSG injection at 2 days pi, and HCG injection at 4 days pi. Ovaries were analyzed at 7 days pi when the C.L formed and when OSE repair was expected to complete. **b, c** In the control group (*Procr-CreER*), ovary whole-mount confocal image with K8 staining indicating a complete repair at the C.L. areas, which can be distinguished by the remaining protrusions (**b**). Section image confirming that the OSE covering C.L area has fully repaired with a continuous K8+ epithelial cell layer (**c**). Scale bars, 100 μm in full ovary view and 20 μm in zoom in view. *n* = 3 mice and 10 images. **d, e** In the ablation group (*Procr-CreER;R26-DTA*), ovary whole-mount confocal image with K8 staining showing the ruptures at the C.L. areas at 7d pi (**d**). Section image also demonstrating gaps in the OSE layer (arrow heads), esp. where C.L areas are (**e**). Scale bars, 100 μm in full ovary view and 20μm in zoom in view. *n* = 3 mice and 10 images

during individual estrus cycles. Targeted ablation of Procr+ cells obliterated the repair process, underlining the biological significance of these cells towards ovulatory rupture repair. Finally, we compared the lineage tracing behavior of Procr+ cells and Lgr5+ cells, providing further evidence for the key role of Procr+ OSE cells as adult progenitor cells.

Our data provide further insight into how Procr+ progenitor cells immediately respond upon OSE rupture. The induction of superovulation results in increased folliculogenesis, and subsequently more ovulatory rupture sites. This increase in the number of rupture sites allowed us to detect the rapid increase of Procr+ cell number at ovulation by FACS analysis. The expansion of Procr+ cells was also visualized at the spot of ovulation, using *Procr-rtTA;TetO-H2B-GFP* mice under DOX administration

throughout the superovulation process. We propose that Procr+ cells symmetrically divide to double the progenitor pool upon ovulatory rupture, because of the following observations. Procr+ cells at ovulation stage display 10.9-fold higher proliferative capacity than Procr− cells. Comparing Procr+ cells at superovulated state and non-stimulated state, their proliferative capacity further enhanced, whereas the proliferation of Procr− cells is not affected by increased ovulatory rupture events. The cue for this amplification might come from particular extracellular signals occurring upon ovulation. The follicular fluid expelled during ovulation consists of Wnts and other potential niche signals[30–33], and may regulate Procr+ progenitor cell expansion.

We cannot formally rule out the possibility that Procr− cells convert to Procr+ cells at the wound edge, owing to the inability

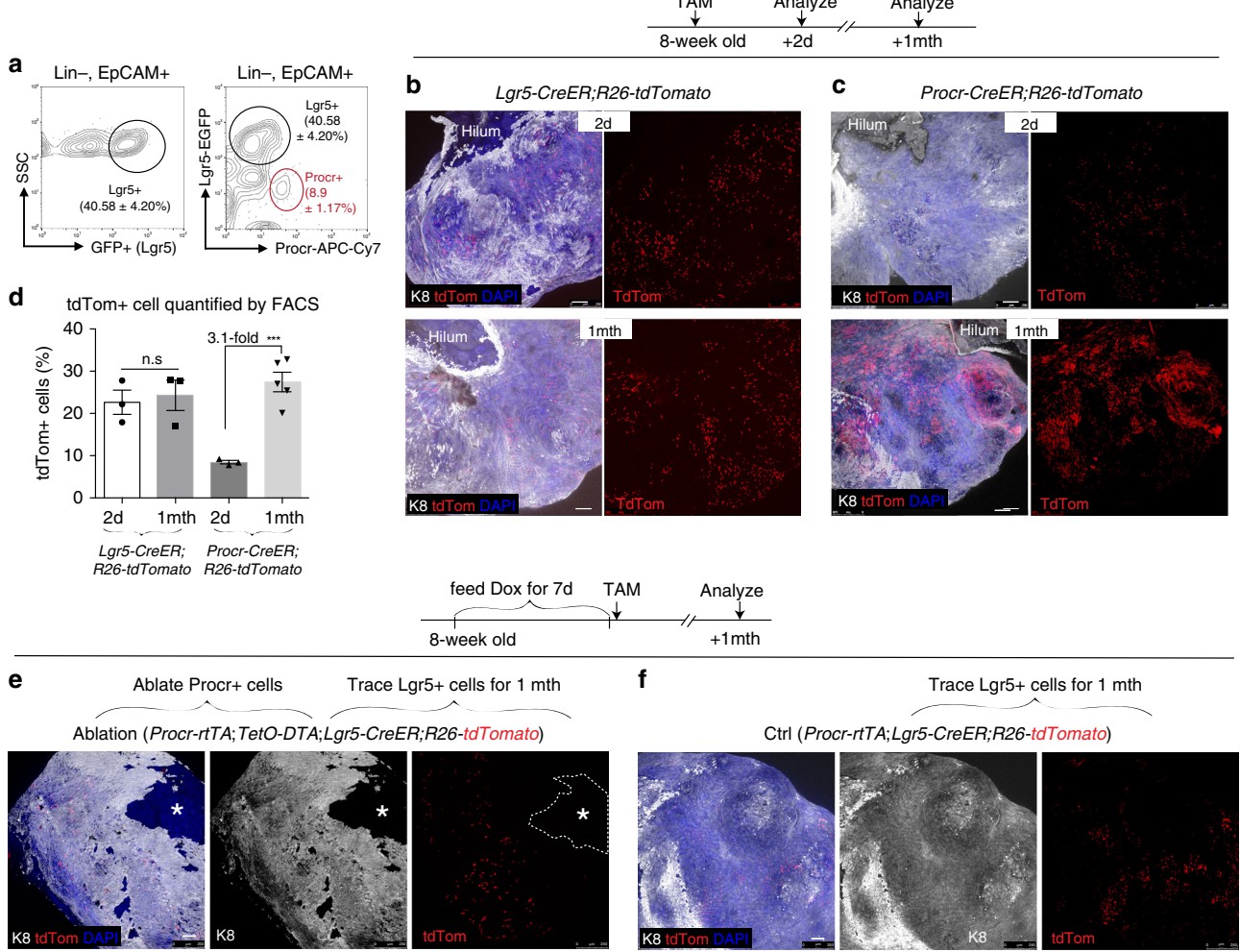

**Fig. 6** Procr+ cell exhibit distinct lineage tracing behaviors compared with Lgr5+ cells. **a** FACS analysis of 8-week-old *Lgr5-CreER-IRES-EGFP* mice ovary showing that the OSE cells, consisting of 40.58 ± 4.20% GFP+ (Lgr5+) cells, and Lgr5+ and Procr+ cells are two distinct OSE cell populations. n = 3 mice. **b**, **c** Lgr5+ cells (**b**) and Procr+ cells (**c**) were traced for 2d and 1 month using (8-week-old) *Lgr5-CreER;R26-tdtomato* mice (**b**) and *Procr-CreER;R26-tdtomato* mice (**c**). Ovary whole-mount confocal images showing the distribution pattern of labeled (tdTom+) cells in the OSE (K8+). Scale bar, 100 μm. n = 3 mice and six ovaries in each time point. **d** FACS quantification of tdTomato+ OSE cells (Lin−, EpCAM+, tdTomato+) after 2d and 1mth tracing, in Lgr5-traced and Procr-traced mouse models. Data are displayed as mean±s.e.m. Unpaired two-tailed t test is used for comparison. ***P < 0.001. n.s, not significant. n = at least three mice in each time point. **e**–**f** Strategy of ablating Procr+ cells and tracing Lgr5+ cells using *Procr-rtTA;TetO-DTA;Lgr5-CreER;R26-tdtomato* mice. Dox was fed into 8-week-old adult mice to ablate Procr+ cells, and 7 days pi, TAM was administered to initiate tracing of Lgr5+ cells. In the ablation group, ovary whole-mount confocal images showing loss of OSE (absence of K8 staining, star). Lgr5+ cell-derived progeny (tdTom+ cells) displayed dispersed pattern, suggesting the loss of OSE injury did not stimulate more robust Lgr5+ cell division (**e**). In control group (*Procr-rtTA;Lgr5-CreER; R26-tdtomato*), ovary whole-mount confocal images with K8 staining indicating the Lgr5+ cell-derived progeny cell pattern in normal homeostasis (**f**). Scale bar, 100 μm. n = 3 mice and six ovaries

of tracing the fate of Procr− cells. However, besides that Procr+ cells are highly proliferative at the right time and right place, several lines of evidence also argue against the conversion of Procr− cells. First, in culture, only Procr+ cells can form colonies, whereas Procr− cells cannot, showing the inability of Procr− cells converting to Procr+ cells. Second, TAM-induced recombination events are most active for a relatively short time period (~2 days)[24–26], and we initiated our lineage tracing protocol of Procr+ cells 4 days before ovulation. Thus, it is most likely that only those Procr+ cells, existing before ovulation, were labeled in our study. Considering one report suggests that TAM may continue to label cells for weeks by Cre-loxP system[34], the behavior of Procr+ cells was further validated by a DOX-mediated pulse/chase experiment. Thus, third, Procr+ cells were marked by nuclear GFP before superovulation, and subsequently gave rise to a substantial number of cells during rupture repair.

These results are consistent with TAM-mediated lineage tracing, highlighting the contribution of Procr+ cells and their descendant cells towards ovulatory repair.

Procr and Lgr5 are both target genes of Wnt signaling[14,27], which has a pivotal role in the development of the ovary[31,35,36]. Previous studies have demonstrated the significance of Lgr5+ OSE cells in early ovarian development[11,12]. In contrast, our study highlights the role of Procr+ OSE cells in adult homeostasis and ovulatory rupture repair. Using *Lgr5-CreER-IRES-EGFP* mouse as a reporter[27], we found that Lgr5+ (GFP+) cells, though comprising a large OSE population in adult (40.6 ± 4.2% of total OSE cells), did not overlap with Procr+ OSE cells. Considering that a modest contribution of Lgr5+ OSE cells in ovulatory rupture repair has been reported previously[12], it is plausible that certain degree of overlapping of Procr+ and Lgr5+ OSE cells can be found if a more sensitive Lgr5 reporter is used.

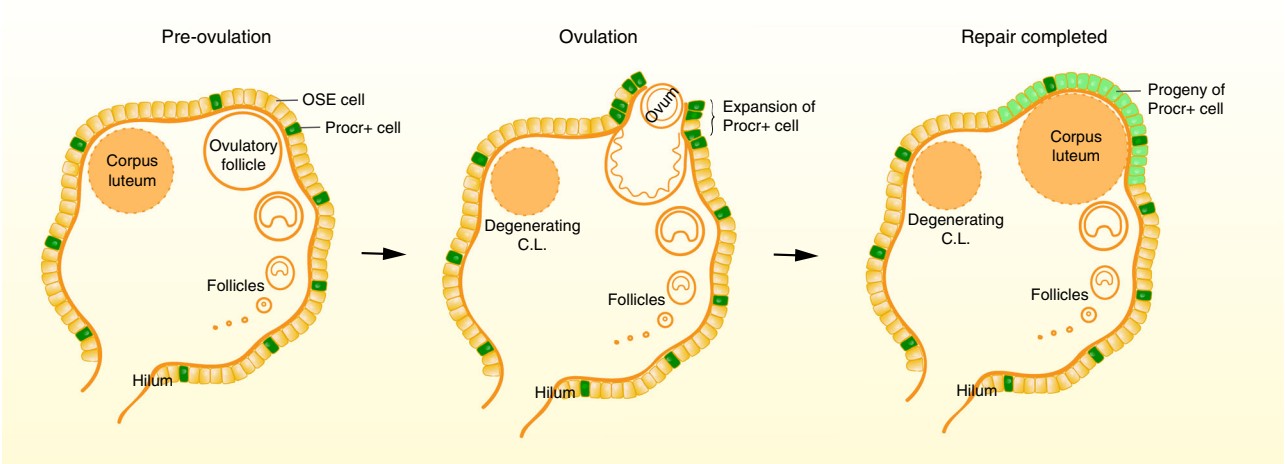

**Fig. 7** Proposed model of Procr+ progenitor cell activity during ovulatory rupture repair. Procr+ progenitor cells (dark green) are distributed in all regions of the ovarian surface (beige). Upon ovulation, Procr+ progenitors near the rupture site expand immediately, facilitating rapid expansion of progeny cells. When repair is completed, Procr+ cells return to their original numbers that are required for their role in maintaining homeostasis. During the process, the progeny of Procr+ progenitors (light green) are the major contributors for replenishing the OSE

In this study, the distribution of Procr+ progenitor cells were thoroughly analyzed by RNA in situ, by using *Procr-rtTA;TetO-H2B-GFP* reporter, and by FACS quantification. During ovulation in poly-ovulatory species such as rodents, as well as in mono-ovulatory species such as humans, the oocyte is released from ovary surface that is not restricted to the hilum region[37]. The widespread distribution of Procr+ progenitor cells across the ovary surface may facilitate rapid and effective re-epithelialization by cells at the wound edge. Our data support the notion that the resident OSE stem cells are not limited to the proximity of previously reported the hilum niche[12].

In this study, we established methods that are advantageous for OSE research in the mouse model. First, we developed a FACS isolation protocol for OSE cells with Lin−, EpCAM+ surface markers. Second, we harnessed the colony-forming capacity of Procr+ progenitor cells, and established a long-term OSE cells culture method. With this ex vivo expansion and long-term culture method, many inherent difficulties owing to insufficient OSE cells for detailed analyses may be resolved. Third, we established an whole-mount confocal imaging technique for ovaries that allows the visualization of the ruptured or mended sites in a complete 3D view. These techniques will help advance our understanding of OSE biology.

In conclusion, our study identifies a population of Procr+ OSE progenitor cells that are used in OSE ovulatory rupture repair and homeostasis in adult mice. The amplification of Procr+ progenitor cells exemplifies a mechanism of how adult stem cells respond upon injury, and adds OSE stem cells to the growing list of adult stem cell lineages, including intestinal and neural stem cells[38,39], in which symmetrical division can be a choice when determining the balance between self-renewal and differentiation[40].

## Methods

**Experiment animals**. *Rosa26^mTmG^*, *Rosa26^tdtomato^*, *Rosa26^DTA^*, *TetO-H2B-GFP*, *TetO-DTA* (The Jackson Laboratories), *Lgr5^EGFP-IRES-CreER^* 27, *Procr^CreER^*14, *Procr^rtTA^*, and C57BL/6 J were used in this study. The following lines were used for in vivo genetic lineage tracing or targeted ablation: *Procr-rtTA;TetO-H2B-GFP* (Dox inducible Procr+ cell reporter, as well as pause/chase for Procr+ cells), *Procr^CreER/+^; Rosa26^mTmG/+^* (Lineage tracing of Procr+ cells with *mTmG* reporter), *Procr^CreER/+^; Rosa26^tdtomato/+^* (Lineage tracing of Procr+ cells with *tdTomato* reporter), *Procr-CreER;Rosa26-DTA* (Ablation of Procr+ cells by DTA), *Lgr5^EGFP-IRES-CreER^;Rosa26-tdTomato* (Lineage tracing of Lgr5+ cells with *tdTomato* reporter), *Procr-rtTA;TetO-DTA;Lgr5-CreER;R26-tdTomato* (Ablation of Procr+

cells and lineage tracing of Lgr5+ cells with *tdTomato* reporter). For lineage tracing experiments, a single intraperitoneal (IP) injection of tamoxifen (TAM; Sigma-Aldrich; catalog #T5648) at a concentration of 2 mg per 25 g body weight diluted in sunflower oil was performed in pubertal and adult mice. For targeted ablation, *Procr^CreER/+^; Rosa26^DTA/+^* female mice between 4-week to 8-week old received two or three intraperitoneal injection of 2 mg per 25 g body weight of tamoxifen (on every second day). For superovulation experiments, 4-week old mice were administered with 10 IU of pregnant mare serum gonadotropin (PMSG) by IP injection, followed by IP injection of 10IU of human chorionic gonadotropin (HCG) 48 h later. For DOX feeding, DOX hyclate (Sigma, catalog #D9891) was dissolved in water at a concentration of 1 mg/ml. Both ovaries were used for research per animal, and the number of mice per experiment was showed in figure legends. Experimental procedures were approved by the Animal Care and Use Committee of Shanghai Institute of Biochemistry and Cell Biology, Chinese Academy of Sciences.

**OSE isolation and flow cytometry**. Ovaries from 7 to 12 weeks old female mice were isolated, and the bursa and oviduct were cleared out under dissect microscope with tined tweezers. The minced ovarian pieces were placed in 10 ml digest buffer (RPMI 1640 (Thermo Fisher, catalog #12633-012) with 5% fetal bovine serum (FBS, Hyclone), 1% penicillin–streptomycin (Thermo Fisher, catalog #15140122), 25 mM HEPES and 300U/ml collagenase IV (Worthington, catalog # LS004189)) and digested at 37 °C, 100 rpm for 1 hour. After lysis of the red blood cells with buffer (Sigma, catalog #R7757) in room temperature for 5 min and centrifugation at 1000 rpm for 5 min, single cells were obtained with 0.05% trypsin-ethylenediaminetetraacetic acid (EDTA) treatment (Thermo Fisher, catalog # 25200056) at 37 °C for 5 min, followed by 0.1 mg/ml DNaseI (Sigma, catalog #D4263) incubation at 37 °C for 5 min with gently pipetting before filtering through 70 μm cell strainers. The single cells were incubated on ice for 25 min with the following antibodies at a dilution of 1:200: fluorescein isothiocyanate (FITC)-conjugated and biotinylated CD31, CD45, TER119 (BD PharMingen, clone MEC 13.3, 30-F11 and TER119; catalog #553372, #553371, # 553672, #55307#, 553080, and # 557915), EpCAM-APC (eBioscience, clone G8.8, catalog 17-5791-82), Procr-PE (eBioscience,clone 1560, catalog #12-2012), Procr-Biotin (eBioscience,clone 1560, catalog # 13-2012-82), Streptavidin-APC-Cy7 (BioLegend, catalog #405208). FITC-conjugated Sca1 (eBioscience, clone # D7, catalog #11-5981-82), Streptavidin-V450 (eBioscience, catalog #48-4317-82). All analysis and sorting were performed using a FACSJazz (Becton Dickinson). The purity of sorted population was routinely checked and ensured to be > 95%.

**OSE cells 3D culture assay**. FACS sorted OSE cells were resuspended with 60 μl 50% growth factor-reduced Matrigel (BD Bioscience, diluted with OSE 3D culture medium) and placed around the rim of a well of a 24-well plate, and allowed to solidify for 25 min at 37 °C in a 5% CO2 incubator before adding 1 ml culture medium. Colonies were grown for 5–8 days. The medium was changed every second days. The culture medium consists of 5% FBS (Sigma), 4 mM L-glutamine (Mediatech), 1 mM sodium pyruvate (Mediatech), 10 ng/ml epidermal growth factor (Sigma), 500 ng/ml hydrocortisone (Sigma), 5 mg/ml insulin (Sigma), 5 mg/ml transferrin (Sigma), 5 ng/ml sodium selenite (Sigma), 0.1 mM MEM non-essential amino acids (Mediatech), $10^{-4}$ M β-mercaptoethanol (Sigma), $10^3$ units per ml leukemia inhibitory factor (Millipore), and Wnt3A (200 ng/ml) or

CHIR99021 (3 μM) in DMEM/F12 (Ham's) medium (Mediatech). For passage of OSE colonies, the colonies were released from Matrigel with dispase at 37 °C in a 5% $CO_2$ incubator for 25 min, followed by digestion with trypsin-EDTA at 37 °C for 5 min to generate single cells. Cells were replated at a split ratio of 1:3 and culture with Fgf2 and CHIR99021.

**Immunohistochemistry.** Ovaries were fixed in cold paraformaldehyde (PFA) for 2 h, washed with PBS for three times and embedded with Optimum Cutting Temperature. Tissue sections were incubated for 20 min in 0.1% Triton-X 100 diluted with PBS (PBST) and blocked for 1 hour using 10% of appropriate serum in PBST. Then sections were incubated with primary antibodies at 4 °C overnight, followed by washing for three times (20 min per time), incubation with secondary antibodies for 2 h at room temperature, followed by washing for three times (20 min per time) and counterstained with DAPI (Thermo Fisher, catalog #D3571). For the samples of isolated cells, cells were fixed in 4% PFA at 4 °C for 15–20 min, followed by the staining protocol described above. For staining of cultured colonies, colonies were fixed on ice for 10 min and washed with PBS, followed by cytospin (Thermo Fisher). The primary antibodies used for immunohistochemistry were rat anti-K8 (1:250, Developmental Hybridoma Bank, catalog #TROMA-I), rabbit anti-K19 (1:500, Abbomax, catalog #602-670), rat anti-EpCAM (1:200, eBioscience, clone G8.8, catalog 17-5791-82), rabbit anti-Laminin (1:500, Sigma, catalog #L9393). At least three times repeats were done per tissue block. Only representative images were shown.

**EdU incorporation.** The proliferation of OSE cells in vivo was measured by 5-ethynyl-29-deoxyuridine (EdU) uptake. In brief, mice were injected with 100 μl EdU (2.5 mg/ml in dimethyl sulfoxide, Life Technologies) for 10 h. Then ovaries were harvested for section, or OSE cells were sorted based on Procr expression. For EdU signal detection, sections or cells were fixed with PFA and pre-treated with 0.5% Triton-X 100 for 30 min Then incubated with the reagents in the Click-iT EdU Alexa Fluor Imaging Kit (Life Technologies), prepared according to the manufacturer's instructions. After EdU signal developing, cells were blocked in blocking buffer for 1 h at room temperature followed by antibody staining and mounted with mounting medium for imaging and quantification.

**Whole mouse ovary immunohistochemistry.** Mouse ovaries that were cleared without bursa and oviduct were fixed with fresh PFA for 15 min at room temperature, and were transferred into 4 ml Eppendorf tubes using a dropper carefully, followed by washing with 0.1% Triton-X100 diluted with PBS (PBST) for three times (20 min for each time). The staining of whole ovaries was continued in the 2 ml Falcon tubes. Ovaries were blocked for 1 hour using 10% of appropriate serum in PBST. Then, the ovaries were incubated with primary antibodies at 4 °C for 48 h upon a transference shaker with 10 rpm, followed by washing for three times (20 min for each time) at room temperature. After washing, the ovaries were incubated with secondary antibodies for 24 h at 4 °C, and counterstained with DAPI (Thermo Fisher, catalog #D3571) upon a transference shaker with 10 rpm, followed by washing for 3 times at room temperature. The ovaries could be stored in PBST at 4 °C for at least 2 weeks. For the whole-mount images, the ovaries were transferred into 35 mm glass bottom dish with PBST over the ovaries. The pictures were captured with inverted Leica TCS SP8 WLL at an ×10 objective, z-stack was ~50–80 layers with 7 μm per layer, and the area was about 1.5 mm x 1.5 mm, which was about 1/6-1/4 of the adult ovary surface, and one ovary could be imaged at different views and stitched to capture the whole ovary. The primary antibody for OSE cells was rat anti-K8 (1:200, Developmental Hybridoma Bank, catalog #TROMA-I).

**RNA in situ.** In situ hybridization was performed using the RNA scope kit (Advanced Cell Diagnostics) following the manufacturer's instructions. *Procr* probes were ordered from Advanced Cell Diagnostics (REF#410321).

**Imaging.** Confocal section images were captured using Leica DM6000 TCS/SP8 laser confocal scanning microscope. Confocal whole-mount images were captured using Leica TCS SP8 WLL.

**RNA isolation and qPCR.** RNA of cells was isolated with Trizol (Thermo Fisher, catalog #15596018) according to the manufacturer's instructions. The cDNA was generated using the SuperScriptIII kit (Thermo Fisher, catalog #18080093). qPCR was carried out on a StepOne Plus (Applied Biosystems). RNA level was normalized to *Gapdh*. Primers used were as follows.

*Procr*-F, CTCTCTGGGAAAACTCCTGACA;
*Procr*-R, CAGGGAGCAGCTAACAGTGA;
*Krt8*-F, AGGATGAGATCAACAAGCGT;
*Krt8*-R, CTTCATGGATCTGCCGGA;
*Krt19*-F, GGGGGGTTCAGTACGCATTGG;
*Krt19*-R, GAGGACGAGGTCACGAAGC.

**Statistical analysis.** Statistical analyses were calculated in GraphPad Prism (Student's *t* test or one-way analysis of variance). For all experiments with error bars, the standard error of measurement (s.e.m.) was calculated to indicate the variation within each experiment.

**Reporting summary.** Further information on research design is available in the Nature Research Reporting Summary linked to this article.

## Data availability

All data are available in this paper or supplementary information, or from the corresponding author upon reasonable request. The source data underlying Figs. 1c, 2f, i, j, 3c, f, i, j, 4b, c, h, i, j, 6d, and Supplementary Figs. 2c, 3e, g, 4f, 5e, and 7c are provided as a Source Data file.

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

## Acknowledgements

We are grateful to Dr. Lindsay Hinck for critical reading of the manuscript. We thank Dr. Chi-chung Hui for helpful discussion. This work is supported by grants from National Natural Science Foundation of China (31830056, 31861163006, 31625020, 31530045, 31661143043 to Y.A.Z., 81873821 to W.L), the Chinese Academy of Sciences (XDB19000000 to Y.A.Z.), and National Key R&D Program of China (2018YFC1002802 to W.L).

## Author contributions

J.W. and Y.A.Z. designed the project and wrote the manuscript. J.W. performed most of the experiments including RNA in situ, FACS analyses, genetic crosses, lineage tracing, targeted cell ablation, and staining. D.W. bred mice for lineage tracing and ablation experiments. W.L. and K.C. helped project design and manuscript editing.

## Competing interests

The authors declare no competing interests.
