## [Peer Review File · Nature Communications]

Reviewers' Comments:

Reviewer #1:

Remarks to the Author:

The Wang et al manuscript which is well written with excellent figures, describes the use of mouse models with conditional and inducible reporters based on Procr1 and Lgr5 expression that track OSE homeostasis and repair during ovulation. The authors convincingly demonstrate that Procr+ progenitors expand during ovulation, and give rise to clonal populations of OSE that replenish the ovulation site. Ablation of Procr+ cells using DT prevents repopulation of the ovulation site. These properties are distinct from the putative progenitors marked by Lgr5 which do not contribute significantly to repair and replenishment of OSE.

Major comments:

The experiments described in this study are well laid out, and their conclusions are definitive. However a few open questions remain.

- 1) Figure 1 shows very limited fields of view of procr staining by RNAish. The authors acknowledge expression in endothelium, which was visible in the medulla in some of the lineage tracing experiment. Given that the FACS was performed on whole ovary digests rather than shorter protocols with intact ovaries, it would be of interest to show Procr expression across the estrus cycle near antral follicles, to rule out neovasculogenesis giving rise to Procr+ Epcam+ cells. It would be useful to quantify by FACS or qPCR the expression or abundance of Procr over the cycle.
- 2) In supplemental figure 6, DT reduces the Procr positive cells from 9% to 2%. Could you provide a later timepoint to show if those numbers ever get restored to normal, as one would expect this should be the case, given the doubling of Procr cells at ovulation, and that there should be 5-6 cycle in one month.
- 3) In supplemental figs4, the labeled OSE appear to undergo hyperplasia at 14 months. Have you tested colony formation efficiency of Procr+ cells compared to Lrg5, or procr- ?

Minor comments:

Although not required for this manuscript, the authors should consider following these elegant studies with a consideration of which estrus specific signals are responsible for initiating this progenitor response.

Line 45: Correct the spelling of keratin and define the abbreviation. The official gene name is Krt.

Line55: The most appropriate term for these cells should be "progenitor" rather than "stem", since pluripotency was not demonstrated in this manuscript.

Line 194: Please add % next to the values.

Line 276: should read as "merger of multiple large clones".

Reviewer #2:

Remarks to the Author:

General comments

This paper provides new and important information about the biology of the ovarian surface epithelium. The major concern is the claim that the increase in the number of Porcr+ cells in association with ovulation is due to a stem cell expansion process rather than simply an increase in the fraction of cells that have activated Procr expression. Fortunately this is an issue that can be readily addressed by co-examination of proliferation markers and identification of capillaries.

Specific comments

1. Line 139 – 140 and lines 152 - 154: The authors suggest that Procr+ cells may grow at an extraordinary rate over 0.5 days and continually expand through day 7. An alternative explanation is that more and more existing cells turn on Procr and, since the half-life of TAM and 4-OH TAM

effect is quite long in mice, the number of GFP+ cells increases in the absence of cell division. The authors cite references 24 – 26 from 1998 – 2006 as evidence that the effect of TAM is short-lived. However, more recent reports indicate otherwise. “These results show that Tm doses commonly used to induce Cre-loxP recombination may continue to label significant numbers of cells for weeks after Tm treatment, possibly confounding the interpretation of time-sensitive studies using Tm-dependent models” (<https://doi.org/10.1371/journal.pone.0033529>). The appearance of more GFP+ cells with time does not necessarily support the conclusion that “Procr+ cells make major contribution to OSE repair after ovulatory rupture”. Co-expression of a cell proliferation marker could help resolve this question.

It is well known that the Procr is involved in the coagulation pathway and is expressed in endothelial cells. Ovarian capillaries can be ruptured when an egg is ovulated. Clinically, this can cause hemorrhagic corpus leuteum and occasionally may require surgery due to hemorrhagic ovarian rupture. While humans ovulate only a single egg, mice ovulate 10-20 eggs at a time. Thus, there is potential for capillary substantial endothelial activation and expression of Procr. This, rather than rapid proliferation, may account in part for the increase in Procr-expressing cells in association with ovulation. Staining for capillary markers may help address this issue.

2. Lines 144-146: The attempt at quantification of the distribution of GFP+ cells within the ovary and at various times after TAM treatment is appreciated. However, it suffers from the non-random selection of control areas of the tissue section. Comparison of GFP+ cell counts in an area of interest relative to number of randomly selected sites in the tissue section would be more convincing.

3. Figure S3e: The non-random selection of sites in which GFP+ cells were counted limits the cogency of this data. Flow cytometric analysis of the fraction of GFP+ cells in the entire NOSE might provide more definitive data.

4. Lines 182 -184: It appears that FCM was used with an anti-Procr antibody to quantify the fraction of total NOSE cells that expressed Procr. As noted for TAM induction, the increase observed may simply be due to more cells starting to express Procr rather than an expansion of cells expressing Procr prior to ovulation.

5. Lines 198-190: What is meant by “peak” GFP level? It appears that all GFP+ cells were identified as “positive” rather than just a fraction of the GFP+ cells. Clarification of how “peak” GFP+ cells were differentiated from non-peak cells is needed.

6. Lines 226-231: As noted above, the data do not strongly support the conclusion that Procr+ cells are all stem cells or that they divide to expand the stem/progenitor pool.

7. The authors make much of the whole mount imaging of the ovary used in this study as an important contribution to the field. However, the description of this technique is very limited and how a large area of the surface of the organ is brought into a single plane adequate for confocal imaging is not clear.

8. The evidence that Procr+ cells are true “stem” cells rather than long-lasting “progenitor” cells is not strong enough to merit the repetitive description of these cells as “stem/progenitor” cells. Procr is expressed in almost every normal human tissue (see below) making its role as an ovarian specific stem/progenitor cell uncertain.

Minor comments

1. The layout of Figure 2 and particularly Figure 3 is confusing. It is difficult to follow the subpanel sequentially.
2. Figures 2i,g and j: Clarify whether these the counts per HPF in an area of 0.04 mm², or are these the counts in the entire 0.04 mm² area?
3. The term "instant upsurge" does not accurately describe the change in Procr+ cells which is clearly not "instant".
4. There are misspelled words throughout the document.

Reviewer #3:

Remarks to the Author:

The manuscript "Pre-expressing stem/progenitor cells are responsible for ovulatory rupture repair of ovarian surface epithelium" is quite interesting and brought some new methodology and important new knowledge for reproductive biologists. The major claim of the manuscript is that there is a population of Procr+ cells in the OSE of mice which is involved in the regeneration of ovaries after ovulation but is different from already known Lgr5+ cells. The new knowledge is also interesting for the reproductive medicine domain and could be implemented also in humans in the future.

Although, there are some issues which need to be addressed before eventual publication.

Here are my comments:

*Authors used the term "stem/progenitor cells" for discovered population of Procr+ cells (Title, Discussion section). Did they apply some other marker to better elucidate the stemness of Procr+ cells from OSE in terms of pluripotency, mesenchymal stem cells or germinal lineage ?

*The FACS procedure was used to quantify the population of Procr+ cells. Did they also sort the population of Procr+ cells from other cells to elucidate their phenotype ? Can authors write something about the phenotype and dimensions of Procr+ cells from OSE ?

*Page 2, Line 48 (Introduction section): Please, precise the "gonadotropic hormone".

*Page 5, Line 125 (Results section): PMSG needs to be explained by whole word: Pregnant mare's serum gonadotropin.

*Page 14 (Methods section): There is no information on the number of animals and their ovaries included into this research and different analyses. Did author research one or both ovaries per animal ? The FACS sorting and whole mouse ovary immunohistochemistry need to be more exactly described.

*Authors need to provide a Table with all types of mice (Procr-rtTA;tetO-H2B-GFP etc.) used in their research with a brief description of characteristics of each to better understand their work.

*It has already been published that stem cells expressing a degree of pluripotency were present in

the OSE of adult ovaries in animal models and in humans but this is not mentioned in the Discussion section.

RE: manuscript NCOMMS-19-14187

All reviewers' comments were constructive and provided invaluable guidance for our resubmission. Based on the comments and suggestions, we have revised the manuscript and figures accordingly. Major improvements are as follows.

- 1) To address whether Procr⁺ cells divide to double the progenitor pool, we investigated the proliferative capacity of Procr⁺ and Procr⁻ OSE cells. We found that upon PMSG+HCG induced superovulation, Procr⁺ cells displayed 11-fold higher EdU incorporation compared to Procr⁻ cells. Comparing Procr⁺ cells at superovulated state and those at non-stimulated state, their proliferative capacity further enhanced, whereas the proliferation of Procr⁻ cells appeared not affected by the increased events of ovulatory rupture (revised Fig. 3).
- 2) To demonstrate the progenitor property of Procr⁺ cells, we included in vitro colony forming analysis. We found that Procr⁺ OSE cells have markedly higher colony formation efficiency compared to total OSE cells, whereas Procr⁻ cells were not able to form colony in culture. Moreover, we found that the maintenance and expansion of Procr⁺ OSE cells are regulated by Wnt signals. Consequently, we established a long-term OSE cell culture method (revised Fig 4).

Reviewer #1 (Remarks to the Author):

The Wang et al manuscript which is well written with excellent figures, describes the use of mouse models with conditional and inducible reporters based on Procr and Lgr5 expression that track OSE homeostasis and repair during ovulation. The authors convincingly demonstrate that Procr⁺ progenitors expand during ovulation, and give rise to clonal populations of OSE that replenish the ovulation site. Ablation of Procr⁺ cells using DT prevents repopulation of the ovulation site. These properties are distinct from the putative progenitors marked by Lgr5 which do not contribute significantly to repair and replenishment of OSE.

We appreciate the reviewer's positive comments.

Major comments:

The experiments described in this study are well laid out, and their conclusions are definitive. However a few open questions remain.

1, Figure 1 shows very limited fields of view of procr staining by RNAish. The authors acknowledge expression in endothelium, which was visible in the medulla in some of the lineage tracing experiment. Given that the FACS was performed on whole ovary digests rather than shorter protocols with intact ovaries, it would be of interest to show Procr expression across the estrus cycle near antral follicles, to rule out neovasculogenesis giving rise to Procr+ Epcam+ cells. It would be useful to quantify by FACS or qPCR the expression or abundance of Procr over the cycle.

We apologize for not have making it clear. Although the FACS was performed on whole ovary digests, we did exclude blood lineage by negative selection of Lin (CD31, TER119, CD45), therefore only quantifying Procr+ cells in Lin-, EpCAM+ OSE cell compartment (Figure 1e).

Following the suggestion, we quantified Procr+ cells across estrus cycle, including proestrus (P), estrus (E), metestrus (M), and diestrus (D) by FACS analysis. No significant difference of Procr+ (Lin-, EpCAM+, Procr+) cells percentage was observed across estrus cycle (see below). We speculate that in contrast to hormone-induced superovulation (Figure 3b, see below), in natural estrus cycle there are not sufficient numbers of eggs, therefore not sufficient ovulatory ruptures that can cause significant change of Procr+ cell percentage when analyzing the whole OSE by FACS.

2, In supplemental figure 6, DT reduces the Procr positive cells from 9% to 2%. Could you provide a later time point to show if those numbers ever get restored to normal, as one would expect this should be the case, given the doubling of Procr cells at ovulation, and that there should be 5-6 cycle in one month.

We appreciate the suggestion. We examined the percentage of Procr+ OSE cells at 4 months after DAT-mediated ablation. The percentage of Procr+ cells got resorted to normal (see below).

3, In supplemental figs4, the labeled OSE appear to undergo hyperplasia at 14 months. Have you tested colony formation efficiency of Procr+ cells compared to Lrg5, or procr-?

We apologize for the misunderstanding it caused. In old Figure S4e, at 14 months, it was a smear caused by sectioning artifact. They have been replaced by better representative pictures in revised figure S4e (see below). The monolayer OSE cells appeared “multilayer”, because it is a Z-stack confocal image from 30 μm-thick sections.

Following the suggestion, we analyzed Procr+ and Procr- cells in colony formation assays, and found that Procr+ cells exhibited 6-fold higher colony formation efficiency compared to total OSE cells, with about one colony formed out of 10 plated Procr+ OSE cells (Fig. 4c). In contrast, Procr- cells were not able to form colony, suggesting that cells with colony-forming ability have been depleted from this group (Fig. 4b-c). In addition, we optimized the colony formation culture protocol and were able to establish a long-term culture of Procr+ cell by serial passage (Fig. 4j-k). We have included these new data in revised Figure 4.

Minor comments:

Although not required for this manuscript, the authors should consider following these elegant studies with a consideration of which estrus specific signals are responsible for initiating this progenitor response.

We appreciate this suggestion and insight. The question raised is interesting, and we will continue to investigate the estrus specific signals that are responsible for this progenitor response.

Line 45: Correct the spelling of keratin and define the abbreviation. The official gene name is Krt.

Noted, and we have corrected the spelling, and have rephrased it as “Krt8 (K8) and Krt19 (K19)” on p.5.

Line55: The most appropriate term for these cells should be “progenitor” rather than “stem”, since pluripotency was not demonstrated in this manuscript.

Following the reviewer’s suggestion, we have changed the “stem/progenitor” to “progenitor” in revision.

Line 194: Please add % next to the values.

We apologize for not making it more clear. In Figure 3, we measured the number of H2B-GFP+ cells in a given area (about 40-50 DAPI+ cells in the scored area, each area is 0.04mm^2), not the percentage. These information was included in the figure legend.

Line 276: should read as “merger of multiple large clones”.

We appreciate the suggestion, and have changed it to “merger of multiple large clones”.

Reviewer #2 (Remarks to the Author):

General comments

This paper provides new and important information about the biology of the ovarian surface epithelium. The major concern is the claim that the increase in the number of Procr+ cells in association with ovulation is due to a stem cell expansion process rather than simply an increase in the fraction of cells that have activated Procr expression. Fortunately this is an issue that can be readily addressed by co-examination of proliferation markers and identification of capillaries.

We appreciate the reviewer's positive comments. Following the suggestion, we have analyzed the proliferation by EdU incorporation, and identified capillaries by ERG staining. These new data have improved the manuscript. We appreciate the advice.

Specific comments

1. Line 139 – 140 and lines 152 - 154: The authors suggest that Procr+ cells may grow at an extraordinary rate over 0.5 days and continually expand through day 7. An alternative explanation is that more and more existing cells turn on Procr and, since the half-life of TAM and 4-OH TAM effect is quite long in mice, the number of GFP+ cells increases in the absence of cell division. The authors cite references 24 – 26 from 1998 – 2006 as evidence that the effect of TAM is short-lived. However, more recent reports indicate otherwise. “These results show that Tm doses commonly used to induce Cre-loxP recombination may continue to label significant numbers of cells for weeks after Tm treatment, possibly confounding the interpretation of time-sensitive studies using Tm-dependent models” (<https://doi.org/10.1371/journal.pone.0033529>). The appearance of more GFP+ cells with time does not necessarily support the conclusion that “Procr+ cells make major contribution to OSE repair after ovulatory rupture”. Co-expression of a cell proliferation marker

could help resolve this question.

We appreciate the reviewer's insight and suggestion. Following the suggestion, we compared the proliferative capacity of Procr⁺ (Lin⁻, EpCAM⁺, Procr⁺) and Procr⁻ (Lin⁻, EpCAM⁺, Procr⁻) cells after 10 hours EdU incorporation. In non-stimulated non-staged ovaries, Procr⁺ cells exhibited 2.9-fold higher EdU incorporation compared to Procr⁻ cells (Fig. 3j). Upon PMSG+HCG induced superovulation, Procr⁺ cells displayed 10.9-fold higher EdU incorporation compared to Procr⁻ cells (Fig. 3j). Comparing Procr⁺ cells at superovulated state and those at non-stimulated state, their proliferative capacity further enhanced, whereas the proliferation of Procr⁻ cells appeared not affected by the increased events of ovulatory rupture (Fig. 3j). Consistently, immunostaining of *Procr-rtTA;TetO-H2B-GFP* ovary sections showed that at the edges of ruptured site, GFP⁺ (Procr⁺) cells have high incidence of EdU staining (Fig. 3k). Collectively, these results suggest that Procr⁺ cells have higher proliferative capacity than Procr⁻ cells, and the proliferation of Procr⁺ cells is further increased upon ovulatory rupture.

It is well known that the Procr is involved in the coagulation pathway and is expressed in endothelial cells. Ovarian capillaries can be ruptured when an egg is ovulated. Clinically, this can cause hemorrhagic corpus leuteum and occasionally may require surgery due to hemorrhagic ovarian rupture. While humans ovulate only a single egg, mice ovulate 10-20 eggs at a time. Thus,

there is potential for capillary substantial endothelial activation and expression of Procr. This, rather than rapid proliferation, may account in part for the increase in Procr-expressing cells in association with ovulation. Staining for capillary markers may help address this issue.

We appreciate the reviewer's suggestion. We apologize that we have not made it more clear. In FACS analysis (Fig. 3b-c), Procr+ cells were quantified in Lin-, EpCAM+ OSE cell compartment, excluding the blood and endothelial lineage contribution (Lin- represents Ter119-, CD45-, CD31-). In immunostaining analysis (Fig. 3e, 3g-h), H2B-GFP+ (Procr+) cells were co-stained with OSE marker K8, and only GFP+, K8+ cells were quantified. Thus the increased Procr+ cells quantified were indeed from OSE contribution. In revised Figure 3a-c, we have adjusted the display of FACS plots, making it more clear that it is the epithelium compartment we analyzed.

Following the reviewer's suggestion, we performed co-staining of ERG (the nucleus of endothelial cells), K8 (OSE) and H2B-GFP (Procr+) staining in *Procr-rtTA;TetO-H2B-GFP* ovaries. We found that in Procr+ OSE cells (H2B-GFP+, K8+, green nucleus surrounding by white cytoplasm, arrowheads) do not overlap with endothelial cells (ERG+, red nucleus, arrows), regardless in Pre-ovulation, Ovulation or Repair completed stages (see below). These results further support that the increased number of Procr+ cells observed after ovulatory rupture are from OSE cell compartment.

2. Lines 144-146: The attempt at quantification of the distribution of GFP+ cells within the ovary and at various times after TAM treatment is appreciated. However, it suffers from the non-random selection of control areas of the tissue section. Comparison of GFP+ cell counts in an area of interest relative to number of randomly selected sites in the tissue section would be more convincing.

We appreciate the reminder. Control areas meant, “Randomly selected non-R.S. (rupture sites)”. It has been changed in the revised figure legend.

3. Figure S3e: The non-random selection of sites in which GFP+ cells were counted limits the cogency of this data. Flow cytometric analysis of the fraction of GFP+ cells in the entire NOSE might provide more definitive data.

Following the suggestion, we included FACS analysis of GFP+ cells in the entire OSE compartment. Indeed labeled cell increased from 7.75 % at 2d pi, to 21.25% at 4wk pi, to 36.21% at 4mth pi (revised Fig. S3f-g).

4. Lines 182 -184: It appears that FCM was used with an anti-Procr antibody to quantify the fraction of total NOSE cells that expressed Procr. As noted for TAM induction, the increase observed may simply be due to more cells starting to express Procr rather than an expansion of cells expressing Procr prior to ovulation.

Please refer to the answer of question #1. We found that Procr+ cells have higher proliferative capacity than Procr- cells, and the proliferation of Procr+ cells is further increased upon ovulatory rupture (Fig. 3j).

5. Lines 198-190: What is meant by “peak” GFP level? It appears that all GFP+ cells were identified as “positive” rather than just a fraction of the GFP+ cells. Clarification of how “peak” GFP+ cells were differentiated from non-peak cells is needed.

In immunostaining analysis, confocal images were stacked to include all potential GFP signals in a given cell, for both Procr+ cells and their progeny cells. At 4d (Preovulation) and 4.5d (Ovulation), the majority of cells were identified as peak cells (Fig.3e, 3g, white arrowheads). In contrast, at 7d (Repair finished), only few cells were identified as peak cells (see Fig. 3h, magnified view 2, white arrowheads), while most GFP+ cells were marked as divided cells (see Fig. 3h, magnified view 1 and 2, blue arrowheads). In

revised manuscript, FACS analysis with quantitative GFP level was added to support the observation. At 4d (Preovulation) and 4.5d (Ovulation), GFP levels were the highest (peak) and more homogenous, whereas at 7d (Repair finished), GFP level had a step-wise decrease (Fig. S5a-b).

6. Lines 226-231: As noted above, the data do not strongly support the conclusion that *Procr*⁺ cells are all stem cells or that they divide to expand the stem/progenitor pool.

Please refer to the answer of question #1. We have included new data to address this following the reviewer's suggestion.

7. The authors make much of the whole mount imaging of the ovary used in this study as an important contribution to the field. However, the description of this technique is very limited and how a large area of the surface of the organ is brought into a single plane adequate for confocal imaging is not clear.

We appreciate the suggestion. A more detailed description has been included in the revised method.

8. The evidence that Procr+ cells are true “stem” cells rather than long-lasting “progenitor” cells is not strong enough to merit the repetitive description of these cells as “stem/progenitor” cells. Procr is expressed in almost every normal human tissue (see below) making its role as an ovarian specific stem/progenitor cell uncertain.

We appreciate the suggestion. We have rephrased it to “progenitor” in the title and text.

Minor comments

1. The layout of Figure 2 and particularly Figure 3 is confusing. It is difficult to follow the subpanel sequentially.

We have arranged the layout of Fig 3.

2. Figures 2i,g and j: Clarify whether these the counts per HPF in an area of 0.04 mm², or are these the counts in the entire 0.04 mm² area?

The counts per HPF in an area of 0.04 mm². More than 40 areas were counted in each group. We have made it more clear in the revised figure legend.

3. The term “instant upsurge” does not accurately describe the change in Procr+ cells which is clearly not “instant”.

We have rephrased it to “rapid increase”.

4. There are misspelled words throughout the document.

Noted and corrected.

Reviewer #3 (Remarks to the Author):

The manuscript “Pre-expressing stem/progenitor cells are responsible for ovulatory rupture repair of ovarian surface epithelium” is quite interesting and brought some new methodology and important new knowledge for reproductive biologists. The major claim of the manuscript is that there is a population of Procr+ cells in the OSE of mice which is involved in the regeneration of ovaries after ovulation but is different from already known Lgr5+ cells. The new knowledge is also interesting for the reproductive medicine domain and could be implemented also in humans in the future.

Although, there are some issues which need to be addressed before eventual publication.

We appreciate the reviewer’s positive comments.

Here are my comments:

*Authors used the term “stem/progenitor cells” for discovered population of Procr+ cells (Title, Discussion section). Did they apply some other marker to better elucidate the stemness of Procr+ cells from OSE in terms of pluripotency, mesenchymal stem cells or germinal lineage?

We have rephrased it to “progenitor” in the title and text.

We did not examine markers/properties of pluripotency, mesenchymal stem cells or germinal lineage, as they are not the most relevant. In current study, we provided functional evidence, including lineage tracing, proliferation, in vitro colony formation, and DTA ablation. In our opinion, these are key data to demonstrate the progenitor property of Procr+ cells.

*The FACS procedure was used to quantify the population of Procr+ cells. Did they also sort the population of Procr+ cells from other cells to elucidate their

phenotype ? Can authors write something about the phenotype and dimensions of Procr+ cells from OSE ?

Following the suggestion, we included in vitro colony formation data. We found that Procr+ OSE cells have more robust colony forming efficiency compared to Procr- OSE cells (Fig. 4a-c); These colonies can be expanded by serial passaging and cultured for long-term cultured in the presence of Wnt3a (Fig. 4j-k).

*Page 2, Line 48 (Introduction section): Please, precise the “gonadotropic hormone”.

Following the suggestion, we have rephrased it as gonadotropin-releasing hormone (GnRH).

*Page 5, Line 125 (Results section): PMSG needs to be explained by whole word: Pregnant mare's serum gonadotropin.

Thanks for the reminder. In revised text and methods, we have included the full names for PMSG and HCG.

*Page 14 (Methods section): There is no information on the number of animals and their ovaries included into this research and different analyses. Did author research one or both ovaries per animal? The FACS sorting and whole mouse ovary immunohistochemistry need to be more exactly described.

We analyzed both ovaries. The number of animals has been shown in the figure legend. Per suggestion, we have included more detailed description for FACS sort and whole mouse ovary immunohistochemistry in the revised methods.

*Authors need to provide a Table with all types of mice (Procr-rtTA;tetO-H2B-GFP etc.) used in their research with a brief description of characteristics of each to better understand their work.

We appreciate the suggestion, and we have provided a table with all mouse lines used.

Mouse line	Function
Procr-rtTA;TetO-H2B-GFP	Dox inducible Procr+ cell reporter, as well as pause/chase for Procr+ cells
Procr-CreER;Rosa26-mTmG	Lineage tracing of Procr+ cells with mTmG reporter
Procr-CreER;Rosa26-DTA	Ablation of Procr+ cells by DTA
Procr-CreER;Rosa26-tdTomato	Lineage tracing of Procr+ cells with tdTomato reporter
Lgr5-EGFP-CreER;Rosa26-tdTomato	Lineage tracing of Lgr5+ cells with tdTomato reporter

Procr-rtTA;TetO-DTA;Lgr5-CreER;R2 6-tdTomato	Ablation of Procr+ cells and lineage tracing of Lgr5+ cells with tdTomato reporter
---	--

*It has already been published that stem cells expressing a degree of pluripotency were present in the OSE of adult ovaries in animal models and in humans but this is not mentioned in the Discussion section.

We appreciate the suggestion. However, we did not examine any pluripotency or germ-line stem cells markers, such as Oct4 and Nanog, as in our opinion, previous reports with pluripotency markers seen in epithelial cells derived from human ovarian follicular fluid are not the most relevant to the current study. The current study investigates markers expressed *in situ* on the OSE layer.

Along this line, in the revision, we included analysis of a classical tissue/adult stem cell marker Sca1, as there has been a report that a subpopulation of Sca1+ OSE cells have enhanced sphere/colony-forming ability (Gamwell et al, *Biology of Reproduction* 2012). We examined the relationship of Sca1+ cells and Procr+ cells in the OSE. Using the updated FACS analysis protocol, we found that Sca1+ cells comprise 11.18±1.51% of OSE cells (Fig. 4d). Interestingly, Sca1+ cells are largely overlapping with Procr+ cells, and all Procr+ cells are Sca1+ (Fig. 4e). These results are in line with the enhanced colony-formation ability seen in Procr+ OSE cells.

Reviewers' Comments:

Reviewer #1:

Remarks to the Author:

The authors have addressed our concerns.

Reviewer #3:

Remarks to the Author:

This manuscript was significantly improved based on my comments and comments of other reviewers.

I suggest this manuscript for publication.